# Purification, Characterization and Bioactivities of Polysaccharides Extracted from Safflower (*Carthamus tinctorius* L.)

**DOI:** 10.3390/molecules28020596

**Published:** 2023-01-06

**Authors:** Qiongqiong Wang, Shiqi Liu, Long Xu, Bin Du, Lijun Song

**Affiliations:** 1College of Food Science and Technology, Hebei Normal University of Science & Technology, Qinhuangdao 066600, China; 2College of Food Science and Technology, Henan Agricultural University, Zhengzhou 450002, China; 3Hebei Key Laboratory of Natural Products Activity Components and Function, Hebei Normal University of Science and Technology, Qinhuangdao 066004, China

**Keywords:** safflower, polysaccharides, structures characterization, physiological activities

## Abstract

Polysaccharides are the main bioactive components in safflower. In this study, safflower polysaccharides (SPs) were extracted by ultrasonic assisted extraction, and four purified safflower polysaccharide fractions (named SSP1, SSP2, SSP3, and SSP4, respectively) were obtained. The physicochemical properties and in vitro physiological activities of the four fractions were investigated. The molecular weights (M_W_) of the SSPs were 38.03 kDa, 43.17 kDa, 54.49 kDa, and 76.92 kDa, respectively. Glucuronic acid, galactose acid, glucose, galactose, and arabinose were the main monosaccharides. The Fourier transform infrared spectroscopy (FT-IR) indicated that the polysaccharides had *α*- and *β*-glycosidic bonds. Nuclear magnetic resonance (NMR) analysis showed that SSP1 had 6 different types of glycosidic bonds, while SSP3 had 8 different types. SSP3 exhibited relatively higher ABTS^+^ scavenging activity, Fe^+3^-reduction activity, and antiproliferative activity. The results will offer a theoretical framework for the use of SPs in the industry of functional foods and medications.

## 1. Introduction

Safflower (*Carthamus tinctorius* L.) is an herbal plant in the family of *Compositae*, which is cultivated extensively in Central Asia [1,2]. In the traditional medicine of many Asia countries, safflower is primarily used to alleviate joint pain, dysmenorrhea, amenorrhea, postpartum abdominal pain, and blood stasis by promoting blood circulation [1]. Modern pharmacological studies have proved that safflower has multiple biological effects, such as antioxidation, antiinflammation, antitumor, and immunomodulatory [2]. These biological effects are closely related to the bioactive components in safflower, such as polysaccharides, pigments, alkaloids, flavonoids, etc. [1,2,3].

Plant-derived polysaccharides are attracting increasing attention in the fields of food, cosmetics, and healthy products. Natural polysaccharides are functional components of many plant materials with remarkable biological activities, such as antitumor, immunomodulatory, antioxidant, and gut microbiota modulation activities [2,4,5]. Safflower polysaccharides (SPs) are dominant and are documented to have many beneficial effects [2]. Li et al. [4] reported that SPs exhibited immunomodulatory and antineoplastic effects by blocking the PI3K/AKT pathway. SPs activated NF-κB via Toll-like receptor 4, therefore inducing cytokine production by macrophages [5]. SPs could also inhibit proliferation and increase apoptosis of HeLa cells by downregulating the PI3K/Akt pathway [6]. In addition, SPs were regarded as a potential alternative drug for the treatment of steroid-induced avascular necrosis of the femoral head [7]. Nevertheless, the functions of SPs are dictated by their chemical structures [2]. To our knowledge, the structural characterization and bioactivities of different purified safflower polysaccharide fractions have been scarcely reported.

In this study, the polysaccharides in safflower were extracted by the ultrasonic assisted extraction and purified by DEAE-52 column and Sephadex G-100 column to obtain four fractions, SSP1, SSP2, SSP3, and SSP4, respectively. The chemical composition, physicochemical properties, structural characteristics, and bioactivities of different purified fractions were further evaluated.

## 2. Results and Discussion

### 2.1. Thermal Characteristic of CSPs

The thermal stabilities of polysaccharides affect their application in the field of food and drug industries. Generally, structural and functional group variations can affect the thermal stability of polysaccharides. The dehydration, decomposition, and oxidation reactions during the heat treatment of polysaccharides can be easily identified by the thermogravimetric curve [8]. Thermo-gravimetric analysis (TGA) was performed to determine the thermal stability of crude safflower polysaccharides (CSPs) (Figure 1). The thermal characteristic curve showed a three-step degradation pattern. The first stage was mainly attributed to a loss of moisture and dehydration. The second stage (180–470 °C) displayed a dramatic decrease in mass (62.56%), which is mainly related to the changes in the depolymerization and decomposition of some functional groups. The above results demonstrated the chemical stability of CSPs in the temperature range of 25 °C to 180 °C.

### 2.2. Isolation and Purification of CSPs

The CSPs solution was separated and purified by DEAE-52 cellulose fast flow, and four independent elution peaks were obtained (Figure 1). These four polysaccharide fractions were named SP1, SP2, SP3, and SP4. The recovery rates of SP1, SP2, SP3, and SP4 fractions based on the amount of crude polysaccharides were 15.7%, 5.5%, 14.3%, and 10.45%, respectively. The four fractions were further purified by Sephadex G-100 column according to the molecular weight (Figure 2). The four final purified fractions were abbreviated as SSP1, SSP2, SSP3, and SSP4, with recovery rates of 62.45%, 63.15%, 61.77%, and 63.89%, respectively.

### 2.3. Chemical Composition

The chemical compositions and zeta potentials of SPs are summarized in Table 1. CSP was found in the highest content of total sugar (66.62%), while SP3 was found in the lowest content (61.77%). The protein content did not show significant differences, which indicated the presence of polysaccharide–protein complexes in SPs [9]. Relatively high contents of uronic acid were found in CSP, SP2, SP3, and SP4 (7.92%, 6.28%, 7.34%, and 7.21%, respectively). It has been demonstrated that the presence of polysaccharide–protein complexes and uronic acid significantly relates to a variety of biological activities [10,11,12].

In general, zeta potential affects the stability of polysaccharides, and polysaccharides with more negative charges exhibit more stable molecular structures [8]. As shown in Table 1, the zeta potentials varied significantly between the SPs. SP2, SP3, and SP4 had high negative charges of −2.78 mV, −8.46 mV, and −16.10 mV, respectively. The result indicated that SP4 was more likely to aggregate and thus, more stable than other SPs.

### 2.4. Structural Characteristics of SSPs

#### 2.4.1. Molecular Weight of SSPs

The HPGPC analysis revealed the homogeneity and molecular weight of SSPs and the results are listed in Table 2. The M_w_ of SSP1, SSP2, SSP3, and SSP4 were about 38.03 kDa, 43.17 kDa, 54.49 kDa, and 76.92 kDa, respectively. The broadband peaks and low PDI values of the four fractions of SSPs (2.35, 3.97, 3.05, and 2.59, respectively) highlighted that they were heterogenic polysaccharides [8].

#### 2.4.2. Monosaccharide Composition of SSPs

The monosaccharide composition of SSPs is shown in Figure 2 and Table 3. All four fractions were composed of the same monosaccharides with different molar ratios. SSP1 was a neutral polysaccharide, and the other three fractions were acidic polysaccharides. The molar ratio of Man, Rib, Rha, GlcA, N-Glc, Glc, Gal, Xyl, Ara, and Fuc in SSP1 were 0.54, 0.19, 0.41, 0.31, 0.11, 4.81, 14.35, 1.05, 13.54 and 0.23, respectively. While Glc, Gal, and Ara were the most abundant monosaccharides of total monosaccharides. For the other three acidic fractions (i.e., SSP2, SSP3, and SSP4), Glc, Gal, and Ara were also the most abundant monosaccharides. Additionally, GlcA and GalA were found in relatively large molar ratios. Among them, SSP4 showed the highest ratio of GlcA and GalA (2.80 and 0.39, respectively), followed by SSP3 (2.32 and 0.25, respectively). This result was slightly different from previous studies. For example, the acidic polysaccharide from *N. tangutorum* Bobr. was composed of Gal, Man, Glc, Ara, and Rha with a molar ratio of 4.63:2.63:2.36:2.19:1 [13]. The predominant monosaccharides for acidic polysaccharides obtained from *Nitraria sibirica* pall were Rha and Gal [14].

The differences in monosaccharide compositions among SSPs might relate to the differences in plant species and states, separation and purification methods, and polysaccharide structures [15,16]. On the other hand, many studies reported that the ratio of GlcA and GalA is significantly related to physiologic activities, such as antioxidant and anti-proliferative activities [16,17]. It is noteworthy that compared to pigments, polyphenols, and flavonoids, findings on the polysaccharides of safflower are still very limited. More studies on the extraction, structure characterization, and biological activity of safflower polysaccharides are warranted.

#### 2.4.3. FT-IR Spectrum of SSPs

FT-IR can be used to identify characteristic functional groups of polysaccharides and analyze their structures [9]. The FT-IR absorption of SSPs is shown in Figure 3 and Table 4. The strong peaks near 3370 cm^−1^ and 2930 cm^−1^ represent the typical peaks in polysaccharides. The broad absorption peak near 3370 cm^−1^ was attributed to the O-H stretching vibration, and the peaks near 2930 cm^−1^ were attributed to the -CH stretching vibrations, including -CH, -CH_2_, and -CH_3_ [18]. The strong band near 1648 cm^−1^ was caused by the vibration of O-H and -COOR [18]. The absorption peaks near 1451 cm^−1^ also indicated the bending vibrations of C-H, C-O, and C=O, indicating the presence of uronic acids [19]. This result was in agreement with the monosaccharide composition. Furthermore, the peak near 1250 cm^−1^ was a characteristic S=O asymmetric stretching vibration of sulfate radicals [18,19]. The absorption signals near 1035 cm^−1^ were attributed to C-O-H and C-O-C, which indicated the presence of pyranose sugars [20]. The areas between 1200 cm^−1^ and 800 cm^−1^ were typical fingerprints of carbohydrates, including C-O-C glycosidic bond vibrations and C-O-H links [21,22]. The signals in the range of 1200–1100 cm^−1^ were caused by the stretching vibration of the glycosidic bond. The band at 868 cm^−1^ confirmed the existence of alpha configurations [23,24]. Moreover, the peaks at 763 cm^−1^ and 674 cm^−1^ might be attributed to the existence of β-configuration of pyranoses [24]. The characteristic absorption around 868 cm^−1^ indicated the possible presence of α-pyranose [16].

#### 2.4.4. NMR Spectrum of SSPs

The detailed structures of complex polysaccharides can be efficiently characterized by NMR, including the α-or β-anomeric configuration, the glycosidic bond types, and sequences of the monosaccharide units [7]. The structure of SSPs were analyzed by 1D NMR (^1^H, ^13^C) and 2D NMR, chemical shift (COSY), heteronuclear multiple-correlation bond (HMBC), and distortionless enhancement by polarization transfer (DEPT).

Figure 4 shows the NMR spectrums of SSP1. In the ^1^H NMR spectrum (Figure 4a), some peaks overlapped in the resonance region of 4.3–4.8 ppm and 4.8–5.5 ppm. The strong signals at 4.52 ppm were attributed to C1 proton signals of Glc. In addition, seven strong peaks (with chemical shifts of 5.13, 4.98, 4.43, 4.37, 5.05, 4.85, and 4.54 ppm), and one weak peak (with a chemical shift of 5.26 ppm) were revealed in the anomeric region. The findings indicated that SSP1 has both α and β configuration glycosidic bonds [25,26].

The ^13^C NMR spectrum (Figure 4b) revealed five strong peaks (with chemical shifts of 109.16, 107.33, 103.01, 103.3, and 99.9 ppm), and three weak peaks (with chemical shifts of 98.84, 104.35, and 99.79 ppm) in the anomeric region. From the HSQC spectrum (Figure 4c) and ^1^H NMR spectrum (Figure 4a), the signals of anomeric hydrogen corresponding to anomeric carbon of SSP1 were obtained, and the chemical shifts were 5.13, 4.98, 4.43, 4.37, 5.05, 4.85, 4.54, and 5.26 ppm. By comparison with the data reported previously, we inferred that the heterotopic carbon (C1)/heterotopic hydrogen (H1) signals of 109.16/5.13, 107.33/4.98, 103.01/4.43, 103.3/4.37, 99.9/5.05, and 98.84/4.85 were assigned as α-1,3-Ara*f*, α-1,3,5-Ara*f*, β-1,3,6-Gal*p*, β-T-Gal*p*, α-1,2,4-Rha*p*, and α-T-Glc*p*, respectively [25,26,27]. Additionally, the types of linkage for both residues G and H were indeterminate because of the overlaps of signals (H-2/C-2 to H-6/C-6) with the other sugar residues (Table 5).

Furthermore, the carbon and hydrogen chemical shift assignments of the major sugar residues in the polysaccharide fractions were obtained. Taking sugar residue A as an example, the chemical shift of the anomeric carbon (C1)/anomeric hydrogen (H1) signal was 109.16/5.13 ppm (Figure 4a–c), and the chemical shift of the H1/H2 signal could be found in the spectrum as 5.13/4.10 ppm (Figure 4d), the chemical shift of C2 associated with H2 was 81.23 ppm (Figure 4c). Similarly, the corresponding hydrocarbon signal chemical shifts on sugar residue A were as follows: C3/H3 (4.02/83.77 ppm), C4/H4 (3.84/76.69 ppm), C5/H5 (3.61, 3.73/61.25 ppm) (Figure 4d–f) [25,26,27,28]. The chemical shift assignments of the main sugar residues of SSP1 are listed in Table 5.

In the same way, the chemical shift assignments of the main sugar residues of SSP3 were obtained according to the NMR results (Figure 5a–f), which are listed in Table 6.

#### 2.4.5. SEM of SSPs

The stereoscopic shape of polysaccharides is closely associated with their physical characteristics, and the structural morphology can be characterized by SEM [29]. The SEM photographs at different magnifications (2000 mag and 20,000 mags) of SPs are shown in Figure 6. The CSP had rough surfaces, showing a porous lamellar structure with some debris and branching structures. This phenomenon suggested that the high-frequency vibration of ultrasound had a significant effect on the surface structure of SPs [29,30]. The fractions of SSP1, SSP2, and SSP4 appeared similar microstructural properties, exhibiting irregular flaky aggregated surface shapes with aggregation status. Thus, they might have larger surface areas. Notably, SSP3 had a smooth and complete structural feature, which was similar to the polysaccharides extracted from *Morinda citrifolia* L. (Noni) [29]. This distinctive structure might endow SSP3 with special physicochemical properties [29,31].

### 2.5. Physiological Activity of SSPs

As shown in Figure 7, the DPPH and ABTS radical scavenging activities, and T-AOC of SPs, increased with the increment of concentration, with a certain concentration dependence.

At the concentration of 0.7 mg/mL, the DPPH·scavenging rates of CSP, SSP1, SSP2, SSP3, and SSP4 were 49.90%, 12.3%, 25.68%, 35.22%, and 32.89%, respectively. The CSP exhibited the highest DPPH·scavenging ability, followed by SP3, and SP4. Meanwhile, the ABTS^+^ scavenging rates of the above SPs were 80.97%, 50.65%, 55.15%, 86.75%, and 52.66%, respectively. The SSP3 showed the strongest scavenging ability against ABTS^+^, followed by CSP, and SSP2. T-AOC assay was used to evaluate the Fe^+3^ reduction activity of SPs. As shown in Figure 7, SSP4 and SSP3 exhibited relatively higher reduction activities, with T-AOC values of 1.35 U/mL and 0.93 U/mL at 1.5 mg/mL. Additionally, the antiproliferative rate against A545 cells of SSPs also showed a dose-dependent relationship. The antiproliferative rates of SSP3 and SSP4 were significantly higher than that of the other three SPs. The antiproliferative rates of the SPs were 38.53%, 21.61%, 29.65%, 45.67%, and 41.52%, respectively.

Our study aimed to screen the optimal polysaccharides fraction by evaluating the physiological activity of SPs. We found that SSP3 and SSP4 exhibited relatively high physiological activities, including ABTS^+^ scavenging activity, Fe^+3^ reducing activity, and antiproliferative activity. Presumably, SSP3 and SSP4 had a high content of GlcA and GalA. It was reported that the antioxidant activities of polysaccharides were significantly affected by their structures and chemical compositions [15,29]. The physiological activities of polysaccharides are positively related to the uronic acids and protein contents, functional groups (such as C=O and OH), and monosaccharide compositions (such as Rha, Glc, GalA, and Ara) [32,33,34]. This documentation well explained the results of this research. All in all, these findings implied that SSP3 and SSP4 served as potential natural antioxidants for the prevention of oxidative stress and associated damages. Further studies are urgently needed on the in vivo physiological activities and their mechanism. Additional studies are necessary to address the needs of polysaccharide manufacturing on an industrial scale.

## 3. Materials and Methods

### 3.1. Plant Materials

The samples of safflower were obtained in October 2020 at Yumin County, Tacheng City, Xinjiang, China (Latitude: 82°99′ *n*, Longitude: 46°20′ E). Eight kg of Safflower was collected from different directions in the planting area. After collection, the samples were cleaned with distilled water, lyophilized, and ground into a powder before being screened through a 60-mesh screen and stored at −18 °C for future determination.

### 3.2. Chemicals

The monosaccharide standards (D-arabinose, L-rhamnose, D-mannose, D-ribose, D-glucose, D-xylose, D-galactose, D-glucuronic acid, and D-galactose acid), human lung cancer cell line A549, DEAE-52 cellulose, deuterium oxide (D_2_O), 1-phenyl-3-methyl-5-pyrazolone (PMP, M70800-100G) dimethyl sulfoxide (DMSO), dialysis bag (8–14 kDa), total antioxidant activity kit (T-AOC), 1,1-diphenyl-2-picrylhydrazyl (DPPH), 2,2′-azino-bis 3-Ethylbenzothia-zoline-6-sulfonic acid (ABTS), total antioxidant capacity kit (T-AOC), and Cell Counting Kit-8 (CCK-8) were provided by Solable Technology Co., Ltd. (Beijing, China). A series of dextran standards (5 k, 50 k, 150 k, 650 k, and 1100 k) were obtained from Aladdin Reagent Int. (Shanghai, China). High-performance liquid chromatography (HPLC) grades of methanol, acetonitrile, and formic acid were purchased from J&K Scientific Ltd. (Shanghai, China). The other chemicals of analytical grade were purchased from Beilian Chemical Co., Ltd. (Tianjin, China).

### 3.3. Extraction of SPs

The sample powder (20 g) was mixed with distilled water (600 mL) and extracted using an ultrasonic extractor (Scientz Inc., Ningbo, China) at 85 °C for 45 min two times. The ultrasonic power and frequency were 360 W and 40 kHz, respectively. After extraction, the obtained supernatant was collected and concentrated to dry mass using a RE-3000 vacuum distillation apparatus (Yarong Instrument Co., Ltd., Shanghai, China). The extract was then precipitated overnight at 4 °C using four volumes of 95% ethanol. The obtained mixture was centrifuged at 5000 rpm for 20 min. Afterward, the precipitation was collected and redissolved with distilled water. Finally, the crude safflower polysaccharides (CSP) were lyophilized and stored at −18 °C for further analysis.

### 3.4. Isolation and Purification of SPs

The CSPs were isolated and purified according to the method described by Ji et al. [35]. Briefly, the Sevag method (chloroform/n-butanol = 4:1, *v/v*) was used to deproteinize the CSP solution. Lipids and pigment were then removed using organic solvents, and the CSP solution was dialyzed for 48 h in a dialysis bag (MWCO 14,000 Da, Union Carbide). Then, the sample was filtered through a DEAE-52 cellulose fast-flow chromatography column (1.6 × 20 cm), which was equilibrated with distilled water. The polysaccharides were eluted with distilled water and various concentrations of NaCl solution (0.05, 0.1, 0.15, and 0.2 mol/L). The eluates were collected by the phenol-sulfuric acid method, and four fractions were obtained and named SP1, SP2, SP3, and SP4. The SPs solution fractions were then concentrated and lyophilized. Sephadex G-100 column (1.650 cm) with distilled water and NaCl solution (0.05, 0.1, 0.15, and 0.2 mol/L NaCl) were employed for further purification. Four eluents were collected and named SSPs1, SSPs2, SSPs3, and SSPs4.

### 3.5. Physicochemical Characteristics Analysis

#### 3.5.1. Chemical Composition

The amounts of total sugar, protein, and uronic acid were determined by the phenol-sulfuric acid method [36] the Bradford method [37], and the m-hydroxy diphenyl method [38].

#### 3.5.2. Zeta-Potential

Samples were dispersed in phosphate buffer (50 mmol/L, pH 7.0) to form a 0.2% polysaccharide solution. The zeta potential was measured at 25 °C using the Zeta-PALS instrument (Zetapals, Brookhaven, USA).

#### 3.5.3. Thermal Properties

Thermogravimetric (TG) and differential scanning calorimetry (DSC) of CSP were tested using a thermogravimetric analyzer (STA 449 F3, Netzsch, Germany). Briefly, samples (2.0 mg) were heated from 25 °C to 700 °C (10 °C/min) in an Al_2_O_3_ crucible with the protection of a nitrogen atmosphere (50 mL/min).

#### 3.5.4. Molecular Weight

The molecular weight (M_w_) was analyzed by high-performance gel permeation chromatography (HPGPC) using a Waters 1515 system equipped with a refractive index detector (1260 RID, Agilent Technologies, CA, USA), and a G4000-swxl column (4.6 mm × 300 mm, Tosoh Biosep, USA). The HPGPC analysis was performed at 30 °C. 10 μL of SSPs (2 mg/mL) was eluted with KH_2_PO_4_ solution (0.1 mol/L) at a flow rate of 1.0 mL/min. Calibration was performed relative to dextran [9].

#### 3.5.5. Monosaccharide Composition

High-performance liquid chromatography with a 1-phenyl-3-methyl-5-pyrazolone (PMP) derivatization method was used to analyze the monosaccharide composition [10]. The SSPs (5.0 mg) were hydrolyzed with trifluoroacetic acid (TFA, 5.0 mL, 3.0 mol/L) at 110 °C for 8 h. Then, the solution was blown dry with nitrogen and dissolved in distilled water (300 μL). Then, the SSPs hydrolysates (250 μL, 1.0 mg/mL) were mixed with NaOH solution (250 μL, 0.6 mol/L) and PMP-methanol solution (500 μL, 0.4 mol/L). Afterward, the solutions were reacted at 70 °C for 60 min in the dark. After cooling for 10 min, HCl solution (100 μL, 0.3 mol/L) was added to terminate the reaction. Then, the chloroform (1.0 mL) was added and the vortex was extracted three times. After centrifugation at 10,000 r/min for 5 min, the supernatant was analyzed by HPLC.

An Ultimate 3000 HPLC system, equipped with a diode array detector and an ultimate C18 column (4.6 mm × 200 mm, 5 μm), was used for determination. The analysis temperature was 30 °C, and the detection wavelength was 250 nm. The mobile phases were 83% of phosphate buffer (pH 6.7, 50 mol/L) and 17% of acetonitrile, at a flow rate of 1.0 mL/min. The qualitative and quantitative analysis of monosaccharides were conducted by the external standard method using monosaccharide mixture standards (2.0 mg/mL).

### 3.6. Structural Characteristics

#### 3.6.1. FT-IR Analysis

According to the KBr disk method to prepare the infrared spectrum for SSPs powder [11], the dried SPs (1.0 mg) was mixed with KBr powder (100 mg), then ground and pressed into pellets. By applying a Nicolet 5700 FT-IR Spectrometer (Thermo Electron, Madison, WI, USA), the FT-IR spectrum of the SPs was collected over the frequency range of 4000–400 cm^−1^.

#### 3.6.2. NMR Analysis

For NMR analysis, SSPs (10.0 mg) were suspended in 500 mL of D_2_O (99.96%) and freeze-dried twice before being dissolved in high-quality D_2_O (500 mL). The ^1^ H NMR spectra were collected by an AVANCEIII 400 MHz spectrometer (Burker, Germany) at 25 °C [11].

#### 3.6.3. SEM Analysis

The ultra-high resolution thermal field emission scanning electron microscope (QUANTA400FEG, FEI Company, USA) was used to investigate the microstructure of the safflower extract residues. The samples were rendered conductive through the deposition of a layer of gold by sputter coating before imaging under a high vacuum at a voltage of 10.0 kV.

### 3.7. Physiological Activity

#### 3.7.1. DPPH Assay

The modified method of Wang et al. [30] was used to determine the DPPH radical scavenging activity. Briefly, 0.2 mL of SSPs solution (0.10, 0.30, 0.50, 0.70, and 0.90 mg/mL) and 0.5 mL of DPPH solution (0.1 mmol/L) were mixed. After reaction for 30 min at 25 °C in the dark, the absorbance was measured at 517 nm. The following formula was used to calculate the activity:(1)$$\mathrm{DPPH}\cdot \;\mathrm{scavenging}\;\mathrm{activity}\;\;(\%)=(A_{0}-A_{i})/A_{0}$$
where A_0_ is the absorbance of the DPPH solution without the sample; A_i_ is the absorbance of the DPPH solution with the sample.

#### 3.7.2. ABTS Assay

The ABTS radical scavenging activity was determined to follow the Wang et al. [15] method with slight modification. Thirty µL of SSPs solution was mixed with 180 µL of working solution (prepared by 7 mmol/L ABTS+ solutions and 2.45 mmol/L K_2_S_2_O_8_ solution) in equal quantities. After reaction for 10 min at 25 °C in the dark, the absorbance of 734 nm was determined. The following formula was used to calculate the activity:(2)$${\mathrm{ABTS}}^{+}\;\;\mathrm{scavenging}\;\mathrm{activity}\;(\%)=(A_{0}-A_{i})/A_{0}$$
where A_0_ is the absorbance of ABTS solution without the sample; A_i_ is the absorbance of ABTS solution with the sample.

#### 3.7.3. Fe^+3^-Reduction Activity

T-AOC kit was used to determine the Fe^+3^-reduction activity according to the operating manual. The T-AOC values were calculated using the calibration curve and expressed as U/mL [39].

#### 3.7.4. Antiproliferative Activity

The antiproliferative activity on A549 cells was determined according to our previous method [39]. Briefly, cells (2 × 10^4^ cells/mL) were seeded into a 96-well plate and incubated for 48 h in a CO_2_ incubator (37 °C, 5% CO_2_). Afterward, 20 μL of SSP s solutions (0.1, 0.5, 0.75, 1.0, and 1.5 mg/mL) were added and incubated for 24 h. Then, CCK-8 (10 μL) was added and incubated for another 24 h. Lastly, the absorbances (450 nm) were recorded. The antiproliferative rate was assumed according to the manufacturer’s protocol.

### 3.8. Statistical Analysis

All analysis was performed in triplicates, and the results were expressed as mean ± standard deviation. The statistics were analyzed using SPSS Software 25.0 by Duncan’s test and analysis of variance (ANOVA), and *p* < 0.05 indicated significant differences.

## 4. Conclusions

In this study, the polysaccharides were extracted from safflower, and four polysaccharide fractions were purified with M_w_ of 38.03, 43.17, 54.49, and 76.92 kDa, respectively. The SPs were composed of Man, Rib, Rha, GlcA, N-Glc, Glc, Gal, Xyl, Ara, and Fuc with different molar ratios. The glycosidic bonds of SSP1 and SSP3 were further analyzed by NMR. SSP1 mainly contained 6 kinds of glycosidic bonds (α-1,3-Ara*f*, α-1,3,5-Ara*f*, β-1,3,6-Gal*p*, β-T-Gal*p*, α-1,2,4-Rha*p,* and α-T-Glc*p*, respectively). SSP1 mainly contained 6 kinds of glycosidic bonds (α-1,3-Araf, α-1,3,5-Araf, β-1,3,6-Galp, β-T-Galp, α-1,2-Rhap, β-1,2-Glcp, β-1,3-Glcp, and α-1,3-Galp, respectively). SSP3 and SSP4 exhibited high ABTS^+^ scavenging activity, Fe^+3^ reduction activity, and antiproliferative activity. It can be concluded that SSP3 showed better overall characteristics, and the findings will provide a theoretical basis for the application of safflower polysaccharides in the food and pharmaceutical fields.

## Figures and Tables

**Figure 1 molecules-28-00596-f001:**
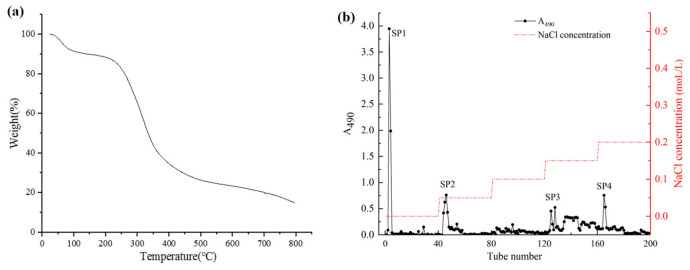
The TGA thermogram (**a**) and elution curve (**b**) of CSP.

**Figure 2 molecules-28-00596-f002:**
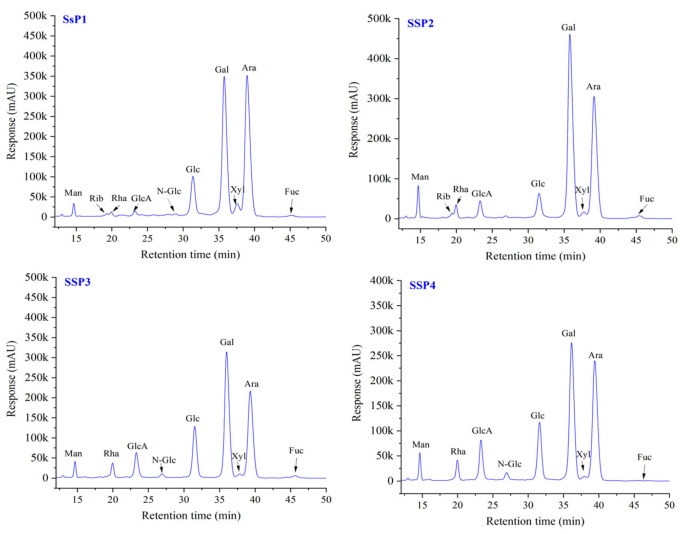
HPLC chromatograms of monosaccharides of SSPs. Abbreviations: Man, mannose; Rib, ribose; Rha, rhamnose; GluA, glucuronic acid; GalA, galactose acid; N-Glu, N-acetyl-amino glucose; Glu, glucose; N-Gal, N-acetyl-amino galactose; Gal, galactos; Xyl, xylose; Ara, arabinose; Fuc, fucose.

**Figure 3 molecules-28-00596-f003:**
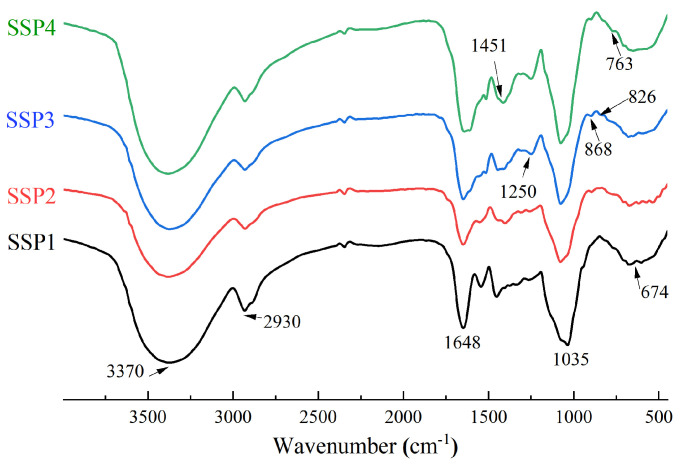
The FT-IR spectrum of SSPs.

**Figure 4 molecules-28-00596-f004:**
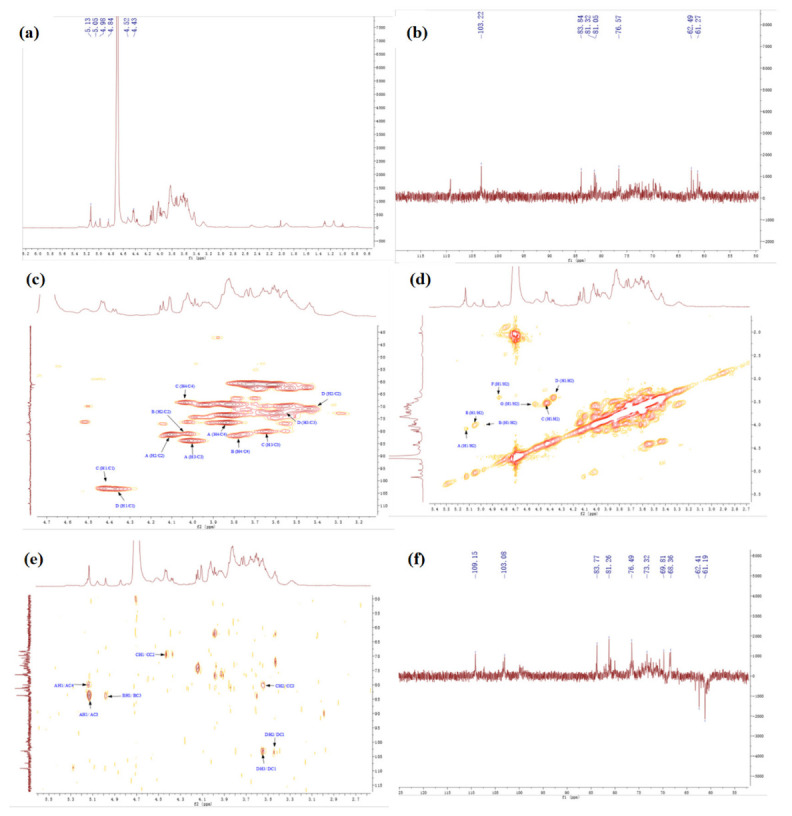
The NMR spectrum of SSP1. (**a**) ^1^H, (**b**) ^13^C, (**c**) HSQC, (**d**) COSY, (**e**) HMBC, (**f**) DEPT.

**Figure 5 molecules-28-00596-f005:**
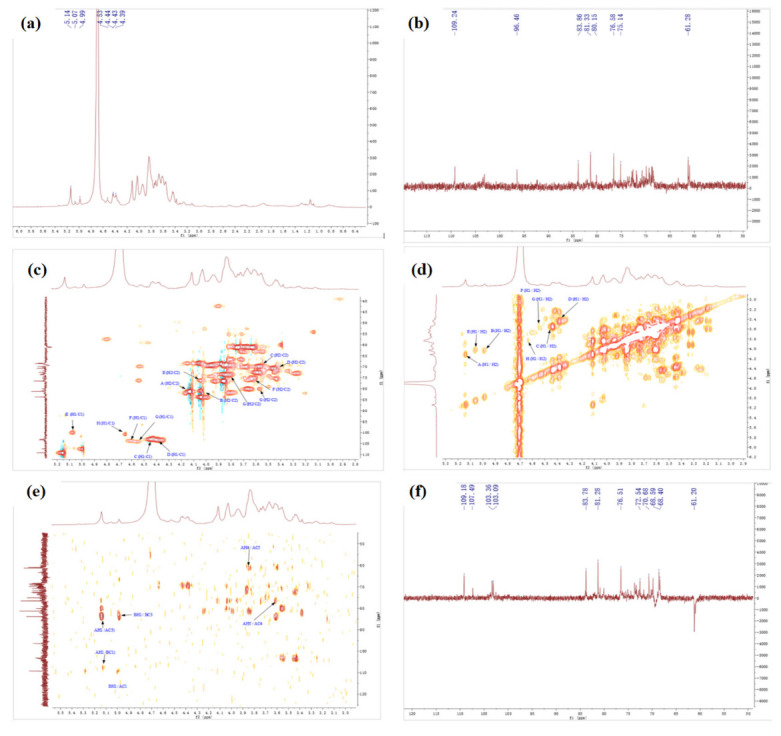
The NMR spectrum of SSP3. (**a**) 1H, (**b**) 13C, (**c**) HSQC, (**d**) COSY, (**e**) HMBC, (**f**) DEPT.

**Figure 6 molecules-28-00596-f006:**
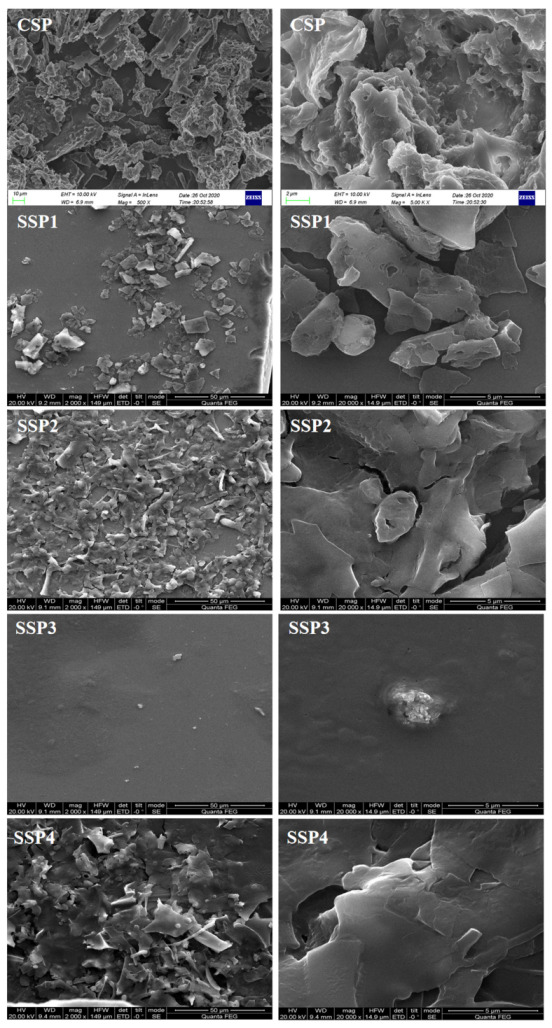
The scanning electron microscopy of CSP and SSPs.

**Figure 7 molecules-28-00596-f007:**
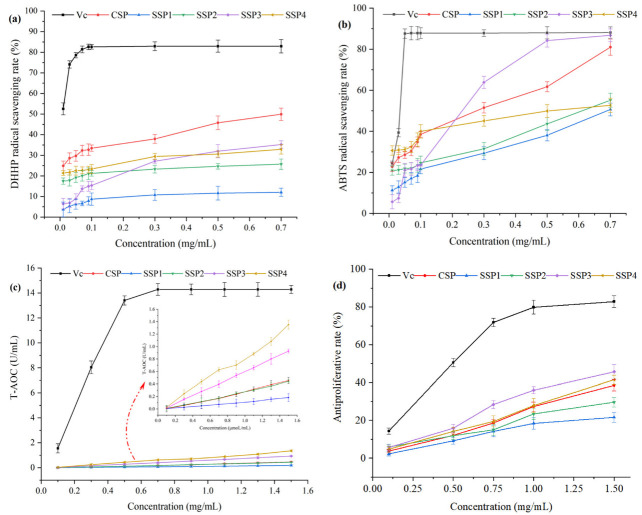
The physiological activities of SSPs. (**a**) DPPH assay, (**b**) ABTS assay, (**c**) T-AOC assay, (**d**) Antiproliferative activity.

**Table 1 molecules-28-00596-t001:** The chemical compositions and zeta potentials of SPs.

	CSP	SP1	SP2	SP3	SP4
Total sugar (%)	66.62 ± 1.66 ^a^	62.45 ± 1.66 ^c^	63.15 ± 1.35 ^c^	61.77 ± 1.19 ^d^	63.89 ± 2.03 ^b^
Protein (%)	5.42 ± 0.41 ^b^	5.19 ± 0.34 ^b^	5.16 ± 0.44 ^b^	5.31 ± 0.30 ^b^	5.89 ± 0.36 ^a^
Uronic acid (%)	7.92 ± 0.31 ^a^	ND	6.28 ± 0.36 ^c^	7.34 ± 0.31 ^b^	7.21 ± 0.27 ^b^
*Zeta* potential	0.51 ± 0.09 ^e^	1.20 ± 0.11 ^d^	−2.78 ± 0.11 ^c^	−8.46 ± 0.24 ^b^	−16.10 ± 0.14 ^a^

a, b, c, d, e: Means with various letters in a row differ significantly (*p* < 0.05).

**Table 2 molecules-28-00596-t002:** Macromolecular of SSPs.

Samples	M_w_ (kDa)	M_n_ (Da)	M_p_ (Da)	PDI (M_w_/M_n_)
SSPs1	38.03	16.16	10.73	2.35
SSPs2	43.17	10.87	12.97	3.97
SSPs3	54.49	17.84	83.34	3.05
SSPs4	76.92	29.69	88.88	2.59

Abbreviations: M_w_, weight average molecular weight; M_n_, number average molecular weight; M_p_, peak average molecular weight; PDI, polydispersity index.

**Table 3 molecules-28-00596-t003:** Monosaccharide composition of SSPs.

	Mannose	Ribose	Rhamnose	Glucuronic Acid	Galactose Acid	N-acetyl-Amino Glucose	Glucose	Galactose	Xylose	Arabinose	Fucose
SSP1	0.54	0.19	0.41	0.310	ND	0.11	4.81	14.35	1.05	13.54	0.23
SSP2	1.29	0.31	1.23	1.35	0.10	ND	3.10	18.53	0.53	11.47	0.21
SSP3	0.73	0.050	1.72	2.32	0.25	ND	6.87	13.90	0.36	9.03	0.20
SSP4	1.02	ND	1.75	2.80	0.39	ND	6.49	12.25	0.27	9.94	0.05

ND, not detected.

**Table 4 molecules-28-00596-t004:** The FT-IR absorption of SSPs.

NO.	SSP1	SSP2	SSP3	SSP4	Primary Vibrations
1	3376.75	3379.26	3373.90	3381.43	υ (OH)
2	2931.38	2930.57	2931.03	2930.93	υ (CH), υ (CH_2_), υ (CH_3_)
4	1648.16	1648.92	1648.35	1642.22	υ(O-H), υ(–COOR)
6	1451.84	1403.37	1445.07	1413.84	υs (C-O), υs (C-H), υs (C=O)
8	1245.90	1242.62	1249.69	1250.23	υs (S=O)
9	1035.27	1078.10	1076.60	1075.95	υ (C-O-H), υ (C-O-C)
10	ND	893.46	868.16	889.32	*α*-pyranose
11	674.51	673.01	677.46	649.07	*β*-configuration

**Table 5 molecules-28-00596-t005:** The ^1^H and ^13^C chemical shifts of SSP1 backbone.

Monosaccharide Residues		Chemical Shift (δ, ppm)					
		C1/H1	C2/H2	C3/H3	C4/H4	C5/H5	C6/H6
A	α-1,3-Ara*f*	109.16	81.23	83.77	76.69	61.25	—
		5.13	4.10	4.02	3.84	3.61/3.73	—
B	α-1,3,5-Ara*f*	107.33	80.98	83.86	81.27	66.66	—
		4.98	4.02	3.99	3.79	3.71/3.80	—
C	β-1,3,6-Gal*p*	103.01	69.79	80.03	68.43	69.18	69.21
		4.43	3.55	3.65	4.04	3.83	3.82/3.94
D	β-T-Gal*p*	103.30	70.85	71.29	73.39	68.65	62.78
		4.37	3.43	3.53	3.84	3.86	3.53/3.69
E	α-1,2,4-Rha*p*	99.90	76.08	74.39	76.88	74.18	16.09
		5.05	4.02	3.93	4.15	3.98	1.32
F	α-T-Glc*p*	98.84	70.96	72.51	72.66	70.99	63.25
		4.85	3.43	3.65	3.3	3.75	3.71
G	β-Glc*p*	104.35	—	—	—	—	—
		4.54	3.55	—	—	—	—
H	α-Man*p*	99.79	—	—	—	—	—
		5.26	—	—	—	—	—

**Table 6 molecules-28-00596-t006:** The ^1^H and ^13^C chemical shifts of SSP3 backbone.

Monosaccharide Residues		Chemical Shift (δ, ppm)					
		C1/H1	C2/H2	C3/H3	C4/H4	C5/H5	C6/H6
A	*α*-1,3-Araf	109.21	81.31	83.87	76.59	61.25	—
		5.14	4.10	4.03	3.84	3.63/3.75	—
B	*α*-1,3,5-Araf	107.39	80.93	83.94	81.91	66.61	—
		4.98	4.03	3.99	3.8	3.71/3.81	—
C	*β-*1,3,6-Galp	103.04	69.87	80.13	68.44	69.17	69.31
		4.43	3.55	3.65	4.05	3.83	3.83/3.94
D	*β*-T-Galp	103.34	70.72	72.55	73.68	68.63	61.01
		4.39	3.44	3.57	3.83	3.87	3.69
E	*α*-1,2-Rhap	99.99	76.34	73.67	68.56	74.23	16.56
		5.06	4.02	3.83	4.14	3.94	1.31
F	*β*-1,2-Glcp	103.81	75.17	74.23	—	—	—
		4.60	3.67	3.93	—	—	—
G	*β*-1,3-Glcp	104.14	72.92	83.01	75.17	—	—
		4.54	3.58	3.85	3.68	—	—
H	*α*-1,3-Galp	100.83	73.65	76.51	—	—	—
		4.65	3.82	3.93	—	—	—

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
