# Peer review of "Purification, Characterization and Bioactivities of Polysaccharides Extracted from Safflower (Carthamus tinctorius L.)"

_molecules, 2023, doi:10.3390/molecules28020596_

Round 1

Reviewer 1 Report

Dear Authors,

My comments are attached.

Regards

Author Response

Dear editor

Thank you very much for your kind comments and constructive revision suggestions on our manuscript “Purification, structural characterization and bioactivities of polysaccharides extracted from Safflower (Carthamus tinctorius L.)” (molecules-2126284). As you and the reviewers suggested, we have edited our manuscript carefully with the help of a language editing company. We have also revised the manuscript according to the reviewers’ comments. We hope the revisions we made (marked in RED in the resubmitted manuscript) can answer you well. The detailed responses to the reviewers are also uploaded. If you have any queries, please don’t hesitate to contact us.

Best regards,

Lijun Song ([email protected]).

December 28, 2022.

Comments and Suggestions for Authors

Reviewer 1

Point 1: Abstract: The abstract is concise and clearly written with a brief explanation of the work that has been carried out. It is adequately structured, consisting of a short introduction, background, methods, results, and obtained conclusion. Line 18: galactose

Response 1: Thanks for your kind comments. We have corrected the spelling mistakes of galactose (in line 19) and checked the spelling of the full manuscript.

Point 2: Introduction: The introduction section is too short and does not provide enough information about the significance of the topic described.

Response 2: Thanks for your kind suggestion. In the revised manuscript, we have added some information in the Introduction section according to your suggestion (in lines 39-43, 53-57). Thanks for your consideration.

Point 3: Materials and Methods: It is not clear how many samples are collected and were the samples representative enough to draw meaningful conclusions.

Response 3: Thanks for your kind suggestion. The safflower was obtained at Yumin County, Tacheng City, Xinjiang, China. The output of safflower in Tacheng City accounts for more than 70% of the total national output. We collected a total of 8 kg of samples from different directions in the planting area. So, the samples were representative enough to draw meaningful conclusions. Additionally, we have added some information about sample collection in the revised manuscript (in lines 238-239).

Point 4: Results and discussion: The results and discussion sections are presented jointly. Obtained results were adequately compared with previous research.

Response 4: Thanks for your kind comment. In the revised paper, we readjusted the order of the three sections (2.1, 2.2, and 2.3 ) to make the article more logical.

Point 5: Conclusion: The main conclusions are given appropriately.

All in all, in my opinion, the presented work does not provide enough scientific soundness and novelty, although many analyses were performed. The extraction and purification procedures are not novel, nor were the methods for performing structural elucidations, as well as antioxidant activities.

Response 5: Thanks for your kind comment. Safflower is a well-known and widely used traditional Chinese medicine that has been verified to have antiinflammatory, antioxidant, and cytotoxic functions. To our knowledge, the research on bioactive compounds in safflower are mostly focused on flavonoids, such as safflower yellow pigment and safflower yellow pigment. However, the current research on Safflower polysaccharides (SPs), especially on the structure and biological activity of purified fractions of SPs is far from sufficient. In this study, we made a detailed study of the physicochemical properties and glycosidic bonds of purified SPs. And the results might provide a theoretical foundation for the application of SPs in different fields.

Frankly speaking, it appeared that we had merely scratched the surface thus far about the research on SPs. Your suggestion will be the main direction of our further research. Further studies are urgently needed to reveal the higher-order structure, structure-bioactivity relationships, as well as bioactivity mechanism of SPs. Additionally, more research needs to be carried out to meet the requirements of polysaccharide production on an industrial scale. And the economy, environmental protection, automation, and comprehensive efficiency should be taken into consideration.

Additionally, we modified the color scheme of Figure 7d to make it consistent with other figures.

Thank you again for your consideration.

Reviewer 2 Report

The aim of the article is very important. Polysaccharides are very important for human health.  Polysaccharides and human health are inextricably linked and intertwined. Polysaccharides are extremely important to sustain our livelihood, soothe our pains, and nurse our wounds. 

Please make all changes in the attached PDF file.

Author Response

Dear editor

Thank you very much for your kind comments and constructive revision suggestions on our manuscript “Purification, structural characterization and bioactivities of polysaccharides extracted from Safflower (Carthamus tinctorius L.)” (molecules-2126284). As you and the reviewers suggested, we have edited our manuscript carefully with the help of a language editing company. We have also revised the manuscript according to the reviewers’ comments. We hope the revisions we made (marked in RED in the resubmitted manuscript) can answer you well. The detailed responses to the reviewers are also uploaded. If you have any queries, please don’t hesitate to contact us.

Best regards,

Lijun Song ([email protected]).

December 29, 2022.

Comments and Suggestions for Authors

Reviewer 2

Point 1: Lines 50-51:Please delete this sentence.

Response: Thanks for your kind suggestion. We have deleted the sentence in the revised manuscript (in Lines 58-59).

Point 2: Lines 57-58:Please change to CSP, SP2, SP3, and SP4 (7.92%, 6.28%, 7.34%, and 7.21%, respectively).

Response: Thanks for your kind suggestion. We have changed the sentence according to your suggestion (in Lines 86-88).

Additionally, we modified the color scheme of Figure 7d to make it consistent with other figures.

Thank you again for your consideration.

Reviewer 3 Report

Dear editor,

The current manuscript entitled Purification, structural characterization and bioactivities of polysaccharides extracted from Safflower (Carthamus tinctorius L.)  explains about the potential use of SPs in the industry of functional foods and medications. It is a well-written manuscript, however, some revisions should be done before acceptance. My specific comments are attached as a pdf file.

Author Response

Dear editor

Thank you very much for your kind comments and constructive revision suggestions on our manuscript “Purification, structural characterization and bioactivities of polysaccharides extracted from Safflower (Carthamus tinctorius L.)” (molecules-2126284). As you and the reviewers suggested, we have edited our manuscript carefully with the help of a language editing company. We hope the revisions we made (marked in RED in the resubmitted manuscript) can answer you well. The detailed responses to the reviewers are also uploaded. If you have any queries, please don’t hesitate to contact us.

Best regards,

Lijun Song ([email protected]).

December 29, 2022.

Comments and Suggestions for Authors

Reviewer 3

Point 1: Lines 2-3: Suggest to change the title to Purification and characterization of polysaccharides from Safflower extract......

Response 1: Thanks for your kind suggestion. According to your suggestion, we have changed the title to Purification, characterization and bioactivities of polysaccharides extracted from Safflower (Carthamus tinctorius L.).

Point 2: Lines 14-24: Abstract section needs to be improved. Line 15: Please write the full name and in what way they are different. Line 16: in vitro change to italic. Lines 23-24: Your conclusion is not clear. You should mention which fraction showed the best activity.

Response 2: Thanks for your kind comments. In the revised manuscript, we carefully revised the Abstract section according to your suggestion (in lines 15-17, 24-25).

Point 3: Lines 50-51: Delete redundant sentences.

Line 54: Please give the full name at the first appearance.

Response 3: Thanks for your kind suggestion. We have deleted the redundant sentences (in lines 58-59).

Additionally, we readjusted the order of the three sections (2.1, 2.2, and 2.3 ) to make the article more logical. The full names of the abbreviations (CSP, SPs) were supplemented (in lines 65, 73-75).

Point 4: Lines 122-123: In Table 3, Need to write the full name in the caption.

Response 4: Thanks for your kind suggestion. We have written the full name of monosaccharides (in lines 130, Table 3) in the revised manuscript.

Point 5: Lines 221: “additional” should be deleted.

Response 5: Thanks for your kind suggestion. In the revised manuscript, we have deleted the redundant word.

Point 6: Line 243: and extracted using......

Response 6: Thanks for your suggestion. We have revised the sentences according to your kind suggestion (in line 257).

Point 7: Line 253: Please use full name at the first appearance.

Response 7: Thanks for your kind suggestion. We have added the full name of CPSs at the first appearance (in line 65). 

Point 7: Lines 364-365: Please mention which fraction showed better overall characteristics.

Response 7: Thanks for your kind suggestion. In the revised manuscript, we have rewritten some conclusions according to your comments (in lines 385-387).

Additionally, we modified the color scheme of Figure 7d to make it consistent with other figures.

Thank you again for your consideration.
